# Development of UPLC-MS/MS Method to Study the Pharmacokinetic Interaction between Sorafenib and Dapagliflozin in Rats

**DOI:** 10.3390/molecules27196190

**Published:** 2022-09-21

**Authors:** Xueru He, Ying Li, Yinling Ma, Yuhao Fu, Xuejiao Xun, Yanjun Cui, Zhanjun Dong

**Affiliations:** 1National Clinical Drug Monitoring Center, Department of Pharmacy, Hebei Province General Center, Shijiazhuang 050051, China; 2Graduate School, Hebei Medical University, Shijiazhuang 050011, China; 3Department of Pharmacy, Hebei General Hospital, Shijiazhuang 050011, China

**Keywords:** UPLC-MS/MS, sorafenib, dapagliflozin, pharmacokinetics, drug–drug interactions

## Abstract

Sorafenib (SOR), an inhibitor of multiple kinases, is a classic targeted drug for advanced hepatocellular carcinoma (HCC) which often coexists with type 2 diabetes mellitus (T2DM). Dapagliflozin (DAPA), a sodium–glucose cotransporter-2 inhibitor (SGLT2i), is widely used in patients with T2DM. Notably, co-administration of SOR with DAPA is common in clinical settings. Uridine diphosphate-glucuronosyltransferase family 1 member A9 (UGT1A9) is involved in the metabolism of SOR and dapagliflozin (DAPA), and SOR is the inhibitor of UGT1A1 and UGT1A9 (in vitro). Therefore, changes in UGT1A9 activity caused by SOR may lead to pharmacokinetic interactions between the two drugs. The objective of the current study was to develop an ultra-performance liquid chromatography-tandem mass spectrometry (UPLC-MS/MS) method for the simultaneous determination of SOR and DAPA in plasma and to evaluate the effect of the co-administration of SOR and DAPA on their individual pharmacokinetic properties and the mechanism involved. The rats were divided into four groups: SOR (100 mg/kg) alone and co-administered with DAPA (1 mg/kg) for seven days, and DAPA (1 mg/kg) alone and co-administered with SOR (100 mg/kg) for seven days. Liquid–liquid extraction (LLE) was performed for plasma sample preparation, and the chromatographic separation was conducted on Waters XSelect HSS T3 column with a gradient elution of 0.1% formic acid and 5 mM ammonium acetate (Phase A) and acetonitrile (Phase B). The levels of Ugt1a7 messenger RNA (mRNA) were determined in rat liver and intestine using quantitative real-time polymerase chain reaction (qRT-PCR). The method was successfully applied to the study of pharmacokinetic interactions. DAPA caused a significant decrease in the maximum plasma concentrations (Cmax) and the area under the plasma concentration–time curves (AUC_0–t_) of SOR by 41.6% and 50.5%, respectively, while the apparent volume of distribution (V_z/F_) and apparent clearance (CL_z/F_) significantly increased 2.85- and 1.98-fold, respectively. When co-administering DAPA with SOR, the AUC_0–t_ and the elimination half-life (t_1/2Z_) of DAPA significantly increased 1.66- and 1.80-fold, respectively, whereas the CL_z/F_ significantly decreased by 40%. Results from qRT-PCR showed that, compared with control, seven days of SOR pretreatment decreased Ugt1a7 expression in both liver and intestine tissue. In contrast, seven days of DAPA pretreatment decreased Ugt1a7 expression only in liver tissue. Therefore, pharmacokinetic interactions exist between long-term use of SOR with DAPA, and UGT1A9 may be the targets mediating the interaction. Active surveillance for the treatment outcomes and adverse reactions are required.

## 1. Introduction

Hepatocellular carcinoma (HCC) is the liver’s most common primary malignant tumor type, with high recurrence and mortality rates [1]. Type 2 diabetes mellitus (T2DM), a group of metabolic diseases, is the most prevalent disease and poses a serious threat to public health [2]. Additionally, T2DM is known to increase the risk of HCC [3,4,5]. Therefore, T2DM is a common comorbidity in HCC patients, and combination therapy is usually recommended for patients suffering from HCC and T2DM, including long-term drug regimens.

Sorafenib (SOR, Figure 1) is the first-line treatment for advanced HCC and exerts inhibitory effects on tumor cell growth and angiogenesis by targeting multiple kinases [6,7]. Following oral administration of SOR, the maximum plasma concentrations are reached in approximately 3 h [8]. The pharmacokinetics of SOR mainly depends on cytochrome P450 (CYP) 3A4, uridine diphosphate-glucuronosyltransferase family 1 member A9 (UGT1A9), and transporters, such as organic anion transporting polypeptide 1B1/3 (OATP1B1/3), P-glycoprotein (P-pg), multidrug-resistance-associated protein 2/3 (MRP2/3), and breast cancer resistance protein (BCRP) [8,9]. Modulation of drug-metabolizing enzymes and transporters activity caused by concomitant medication or pathophysiologies may result in changes in the pharmacokinetics of SOR. These changes may lead to increased or decreased plasma levels of SOR and affect its efficacy [10]. The exposure of SOR may also be affected by enterohepatic recycling as SOR glucuronide is degraded by intestinal β-glucuronidase to become its parent formation again and transported into the blood [11]. In fact, numerous pharmacokinetic interaction studies of SOR involving the CYP3A4, UGT1A9, and transporters have been reported [12,13,14]. More importantly, according to in vitro studies, SOR is one of the most powerful inhibitors of human UGT enzymes identified to date [15]. Previous research reported that SOR elevated the exposure of irinotecan (UGT1A1 substrate) and its active metabolite SN-38 [15,16]. The inhibition of UGT1A1 by SOR also leads to hyperbilirubinemia in patients treated with the drug [17]. However, the influence of SOR on the pharmacokinetics of the UGT1A9 substrate remains unknown, and drug–drug interactions (DDIs) involving UGT1A1 or UGT1A9 are often overlooked in clinical practice.

Dapagliflozin (DAPA, Figure 1), a widely used oral hypoglycemic drug, selectively inhibits the sodium–glucose cotransporter-2 located in the proximal convoluted tubule to block the reabsorption of filtered glucose and promotes its excretion in the urine [18]. It also can lower blood pressure, reduce body weight, and protect the heart and kidney while lowering glucose, supporting its recent approval for chronic kidney disease and heart failure in those with or without diabetes [19]. Currently, some studies [20,21,22] investigated the role of DAPA in reducing fat, decreasing inflammation, and attenuating liver fibrosis, possibly preventing the development of liver tumors or improving early HCC. DAPA is primarily metabolized to inactive metabolite dapagliflozin 3-O-glucuronide via UGT1A9 in the liver and kidneys and then excreted out of the body in the urine. However, it is metabolized less by CYP450 (<10%) [18,23]. One study [24] showed that mefenamic acid (a strong UGT1A9 inhibitor) increased the exposure of DAPA by 51%, and rifampin (a pleiotropic enzyme inducer) reduced its exposure by 22%. In addition, another study [25] reported that DAPA was a moderate UGT1A1 and UGT1A9 inhibitor. Still, the authors ruled out DAPA as a perpetrator of DDIs arising from the inhibition of UGT enzymes based on the in vitro–in vivo extrapolation. However, according to the pharmacokinetic characteristics of DAPA, there is a potential risk for drug interactions with inhibitors and inducers of UGT1A9.

Considering the more favorable therapeutic effects of SOR and DAPA for HCC and T2DM, SOR and DAPA may be co-prescribed in the clinical management for HCC patients co-afflicted with T2DM. Since both drugs are substrates of UGT1A9 and both can inhibit enzymatic activity to different extents, we speculate that pharmacokinetic interactions between the two drugs may exist. However, to our knowledge, there is no information on pharmacokinetic interactions between the two drugs. When given together, it is necessary to characterize the pharmacokinetics of SOR and DAPA.

Several methods have been developed to determine the blood concentration of SOR [26,27,28,29]. However, these methods presented some limitations regarding a narrow linearity range [26,27], a lengthy analysis time [27], and a cumbersome sample preparation [28]. There was a validated method for therapeutic drug monitoring of SOR with a wide linear range and enough sensitivity. However, the analysis time was too long to be used for the analysis of large numbers of samples [29]. Similarly, several methods can be used to determine the blood concentration of DAPA [30,31,32,33]. Some of these methods, however, had a restricted linear range, although a rapid runtime [30,31]. Additionally, some methods needed a high volume of plasma which was difficult to obtain in rats and was unfriendly [30,32]. There was a developed method for determining DAPA blood concentrations in rat plasma with a suitable linear range, but unfortunately, the required analysis time was too long [33]. Therefore, the above methods have characteristics and limitations, but none are suitable for the simultaneous determination of SOR and DAPA in plasma.

Considering the above information, the primary purpose of this study was to develop a rapid and sensitive UPLC-MS/MS method for the simultaneous determination of SOR and DAPA in plasma and to investigate the pharmacokinetic interactions between SOR and DAPA in rats, as well as the changes in the mRNA expression levels of Ugt1a7 (alternative expression of UGT1A9 in rats [34,35]) in the rat liver and intestines.

## 2. Results

### 2.1. Method Validation

#### 2.1.1. Selectivity

Typical chromatograms of SOR, SOR-d_3_, DAPA, and ^2^H_4_-CAPA in blank plasma (Ⅰ), blank plasma spiked with analytes at LLOQ and IS (Ⅱ), and actual plasma samples of rats after oral administration of SOR and DAPA (Ⅲ) are presented in Figure 2. The retention times for SOR, SOR-d_3_, DAPA, and ^2^H_4_-CAPA were 1.98, 1.98, 1.21, and 1.48 min, respectively. There was no apparent interference from the endogenous substances at the retention times of the analytes and IS.

#### 2.1.2. Calibration Curve and LLOQ

Linear regression analysis was used to construct the calibration curves over the concentration ranges of 5–5000 ng/mL for SOR and 5–2000 ng/mL for DAPA, respectively. The typical calibration curves were Y = 0.00459 X + 0.0108 (r = 0.999) for SOR and Y = 0.0054 X + 0.0074 (r = 0.999) for DAPA. The precision (RSD) and accuracy (RE) of the actual concentration at all points on the standard curves were less than 15% of the standard concentration, including the LLOQ, which met the specific criteria.

#### 2.1.3. Precision and Accuracy

The intra- and inter-day precision and accuracy values are summarized in Table 1. The intra-day and inter-day precision ranged from 2.04% to 5.28% and from 4.01% to 5.46%, respectively, for SOR and from 1.75% to 5.14%, and from 3.47% to 6.26%, respectively for DAPA. The intra-day and inter-day accuracy fell in ranges from 0.98% to 3.47% and from −0.12% to 3.89%, respectively, for SOR and from 3.06% to 3.67% and from 3.47% to 6.26%, respectively, for DAPA. The precision and accuracy of the method met the requirements of validation.

#### 2.1.4. Matrix Effects and Extraction Recovery

The matrix effects for SOR and DAPA ranged from 96.97% to 100.78% and from 101.33% to 106.20%, respectively (Table 2). The RSD of the IS-normalized matrix effects did not exceed 9.08% and 5.28% for SOR and DAPA, respectively, and the influence of the biological matrix on the analyte’s response was negligent. Additionally, the extraction recovery for SOR and DAPA in plasma samples at three concentration levels was greater than 97.29% and 91.80%, respectively (Table 3). Furthermore, all analytes achieved satisfactory recoveries, demonstrating that liquid–liquid extraction in this study was excellently able to extract SOR and DAPA.

#### 2.1.5. Stability

All results of the stability tests for SOR and DAPA in rat plasma under different storage and processing conditions are summarized in Table 3. As seen by these results, the SOR and DAPA were stable at the autosampler for 6 h, at room temperature for 4 h, at −80 °C for 30 days, three cycles of freezing and thawing (−80 °C to room temperature), and the RE and RSD values were within an acceptable range of ±15%.

### 2.2. Pharmacokinetic Interactions between SOR and DAPA

#### 2.2.1. Effect of DAPA on SOR Pharmacokinetics

The mean plasma concentration–time profile of SOR after oral administration alone and in combination with multiple-doses DAPA is shown in Figure 3A. The main pharmacokinetic parameters of SOR are summarized in Table 4. The multiple doses of DAPA significantly decreased C_max_, AUC_0–t_, and AUC_0–∞_ of SOR by 41.6%, 50.5%, and 46.2%, respectively, compared with the control group, while the V_z/F_ and CL_z/F_ were significantly increased by 2.85- and 1.98-fold, respectively. In addition, the T_max_ for SOR was also comparable: 1.25-fold shorter (6 h) than that of the control group (7.5 h). However, there were no statistically significant changes for the t_1/2z_, MRT_0–t_, and MRT_0–∞_ between the groups. Moreover, after 72 h of SOR gavage, the pharmacokinetic curves showed a second absorption peak as the concentration of SOR in rats increased again.

#### 2.2.2. Effect of SOR on DAPA Pharmacokinetics

The mean plasma concentration–time profile of DAPA after oral administration alone and in combination with multiple-doses SOR is shown in Figure 3B. The main pharmacokinetic parameters of DAPA are displayed in Table 4. The AUC_0–t_ and AUC_0–∞_ for DAPA significantly increased 1.66- and 1.80-fold, respectively, when DAPA was co-administered with multiple doses of SOR. In addition, the T_max_, t_1/2z_, MRT_0–t_, and MRT_0–∞_ of DAPA significantly elevated by 1.43-, 1.80-, and 1.48-, 1.80-fold, respectively, while the CL_z/F_ of DAPA significantly decreased by 40% compared with the control group. Other pharmacokinetic parameters of DAPA, including C_max_, T_max_, and V_z/F_, revealed no significant difference between the groups.

### 2.3. mRNA Expression in the Liver and Intestines

To gain insight into the possible mechanism of pharmacokinetic interactions between SOR and DAPA via the drug-metabolizing enzyme, we determined the mRNA expression level of Ugt1a7 in the liver and intestines of rats. These results are shown in Figure 4. The effect of DAPA on the mRNA expression of Ugt1a7 revealed that DAPA administered to rats for seven consecutive days significantly inhibited the mRNA expression of Ugt1a7 in the liver by 59.31% but had no significant effect on the expression of Ugt1a7 in the intestines (Figure 4A). Treated with multiple doses of SOR, the mRNA expression of Ugt1a7 in the liver and intestines of rats was approximately downregulated by 32.50% and 32.09%, respectively (Figure 4B).

## 3. Discussion

HCC and T2DM are two chronic diseases with increasing prevalence, and T2DM is a significant risk factor for HCC. It is well acknowledged that appropriate management of coexisting conditions, including T2DM, can improve a patient’s quality of life with HCC. Thus, patients inevitably receive long-term and numerous pharmacological treatments. However, combining multiple drugs may alter the drug profiles on pharmacodynamics and pharmacokinetics attributed to drug interactions. In a comprehensive evaluation [36], the prevalence of DDIs was 88.1% in cancer patients, which increased with the number of comorbidities and drugs used. Consequently, it diminishes the patient’s quality of life by reducing efficacy or increasing the drug-related adverse effects. For example, the previous findings [37] showed that patients with HCC and T2DM receiving the sorafenib–metformin combination presented poorer progression-free survival (PFS) and overall survival (OS) than those administered SOR alone, attributable to increased tumor aggressiveness and resistance to SOR. Another study [14] showed that metformin decreased by a halved exposure to SOR, which may lead to unsatisfactory responses to oncological treatment. Therefore, it is imperative to identify the potential DDIs to reduce the risks of unexpected outcomes.

SOR is the first systemic treatment for advanced HCC. Its antitumor efficacy and adverse effects such as fatigue, diarrhea, hypertension, and hand and foot skin reaction positively correlate with plasma concentration [6,38]. DAPA has become the first-line hypoglycemic agent in addition to metformin in certain patients with T2DM. Similarly, the pharmacodynamic effect is driven by the AUC of DAPA [39]. Moreover, an increase in urinary glucose excretion is dose-dependent and associated with urinary tract infections [23]. However, UGT1A9 is involved in their primary metabolism, and SOR is a potent inhibitor for UGT1A9. Co-administration of SOR and DAPA changes the pharmacokinetic behavior of either drug due to DDIs that may result in complicated outcomes, such as an increase in severe adverse reactions, lack of efficacy, or tolerability issues.

This study simultaneously developed and validated a simple and reliable method for the determination of SOR and DAPA in rat plasma. Liquid–liquid extraction was conducted to enrich cleaner analytes and reduce the interference of substrates. On the other hand, it protects the column well against depletion compared with protein precipitation. Methyl tertbutyl ether acted as a favorable extractant and obtained excellent recovery for poorly water-soluble SOR (LogP3.8) [40] compared with ethyl acetate, whose polarity was stronger than methyl tertbutyl ether. This was also the case for DAPA. Both positive and negative ionization modes of DAPA were detected in this method. The ammoniated adduct ions [M + NH_4_]^+^ of DAPA were found to provide intense signals in positive mode compared with protonated ions [M + H]^+^. It was observed that the mobile phase condition significantly affected the type of adduct formation and its intensity [41], and 5mM ammonium acetate was used in phase A. In addition, the addition of 0.1% formic acid resulted in sharp and symmetrical peaks for analytes.

The method was successfully applied to investigate pharmacokinetic interactions between SOR and DAPA in rats. The dose of 100 mg/kg for SOR used in our study was selected from the previous studies [13,14], and DAPA was administered at 1 mg/kg, which was selected by converting recommended doses for patients in practice to animal doses [42]. In addition, UGT1A9 is functional in humans, whereas it is a pseudogene in rats, and Ugt1a7 compensated for the functions of UGT1A9 in rats [34,35]. For this reason, we measured the mRNA expression of Ugt1a7 in rat liver and intestines to assess its role in the pharmacokinetic interactions between SOR and DAPA.

We found that the co-administration of SOR with multiple doses of DAPA could lead to significant decreases in AUC_0–t_, AUC_0–∞_, and C_max_ of SOR and significant increases in V_z_ and CL_z/F_. The present findings are not consistent with our original assumption. We speculate that this may be related to the enterohepatic circulation effect of SOR. Glucuronide of SOR, produced through glucuronidation of the parent compound by UGT1A9, plays a role in the enterohepatic recycling of SOR. Namely, the plasma exposure of SOR may be affected by enterohepatic recycling as sorafenib glucuronide is degraded by intestinal β-glucuronidase to become its parent formation again and transported by OATP-1B1 into the blood [11,43]. By inhibiting Ugt1a7, DAPA decreased the formation of SOR-glucuronide involved in the enterohepatic circulation and resulted in lower blood concentrations and exposure to SOR. Such a change in pharmacokinetics is consistent with the effect of metformin on the pharmacokinetics of SOR, in which metformin can block the enterohepatic circulation of SOR [14], thereby supporting our hypothesis. Furthermore, previous studies [44,45] demonstrated that DAPA treatment might subtly alter intestinal microbiota composition in type 2 diabetic mice/rat models. Additionally, co-administration of neomycin interferes with the enterohepatic recycling of SOR, resulting in decreased SOR exposure by 54% [46]. Thus, we also presumed that in rats treated with seven-day DAPA, the halved exposure to SOR was observed by interfering with microorganisms with glucuronidase activity, which is consistent with the effect of neomycin on SOR. On the other hand, inhibition of the sorafenib glucuronide transport in hapantotypes by OATP1B1 might decrease the exposure to SOR [43]. However, no study has reported on the inhibition of OATP1B1 by DAPA. In addition, a diminished glucuronidation pathway of SOR may also compensatorily enhance the oxidative metabolic pathway mediated by CYP3A4, causing decreased exposure of SOR in rats. Also, SOR and DAPA are both high protein binding drugs and may compete for the same binding sites, leading to the elevated clearance and apparent volume of distribution of SOR. In summary, the changes in pharmacokinetic parameters of SOR may be explained by a combination of the reasons mentioned above, and many hypotheses require further research. The halved exposure to SOR was almost reduced, and the response to oncological treatment may be dramatically diminished, even leading to resistance to SOR. On the other hand, tissue distribution of SOR increased, making it easier to penetrate and accumulate in tissues, and causing increased toxicity involving the skin, gastrointestinal tract, etc. Nevertheless, it cannot be rejected that additional antitumor effects may be obtained when SOR accumulates in the liver at higher concentrations.

DAPA co-administration with SOR for multiple doses increased the AUC of DAPA. Additionally, a significant decrease in CL_z/F_ and an obvious prolongation of the t_1/2z_ were observed. DAPA is predominantly metabolized by UGT1A9, and the production of glucuronide of DAPA is directly eliminated via renal excretion, without enterohepatic circulation [23]. Rifampin and mefenamic acid, two potential UGT1A9 modulators, changed the total exposure and urinary glucose excretion of DAPA in healthy subjects, as reported by Kasichayanula et al. [24]. Therefore, these changes in the pharmacokinetics of DAPA might be influenced by working through UGT1A9. SOR is a substrate of UGT1A9 and one of the most potent UGT inhibitors in vitro [15]. In our current study, the mRNA levels of Ugt1a7 in the liver and intestines were significantly inhibited by multiple doses of SOR, which was consistent with the findings from in vitro. After multiple administrations of SOR, the metabolic pathway of DAPA in rats via glucuronidation was inhibited, leading to an increase in its exposure and a reduction in its clearance. Correspondingly, the previous study showed the inhibitory effect of SOR on the tapentadol glucuronidation process, in which the C_max_ and AUC_0–t_ for tapentadol increased significantly [13]. In addition, SOR can also inhibit the metabolism of DAPA by competing with DAPA for the same metabolic enzymes to increase its exposure. The results of pharmacokinetic interactions indicate that increased exposure to DAPA may be associated with better glycemic control. However, when combining DAPA with SOR, active surveillance to prevent potential adverse drug of DAPA is essential. It is well known that SGLT2i seem to undergo the same metabolic pathways mediated by UGT1A [47], so we hypothesize that the exposure of other SGLT2i may also change in pharmacokinetic when they are prescribed together with SOR. Some studies have reported that SOR inhibited UGT1A1 more strongly than UGT1A9 [15,17]. Interestingly, it seems that the inhibition of UGT1A9 was also potent after long-term use of SOR in our experiments. Therefore, when SOR is combined with the drugs metabolized by UGT1A9 in HCC patients, physicians and pharmacists should pay special attention to drug interactions mediated by this inhibitory effect.

This study investigated the pharmacokinetic interactions between SOR and DAPA. However, some limitations should be noted for interpreting this result properly. One is that disease states may influence drug disposition. This study did not examine the drug interaction using an animal model with HCC or T2DM. Additionally, species differences exist between rats and humans, and precautions should be noted when extrapolating these data to humans. Thus, further studies in clinical settings are warranted.

## 4. Materials and Methods

### 4.1. Chemicals and Regents

SOR (purity 99.8%, Lot ZZS-20-638-G3), Sorafenib-d_3_ (purity 99.5%, Lot ZZS-20-X261-A1), and ^2^H_4_-Canagliflozin (purity 98%, Lot 21J167-D1) were purchased from Shanghai Zhen Zhun Biotechnology Co., Ltd. (Shanghai, China). DAPA (purity ≥99%, Lot K1704045) was obtained from Shanghai Aladdin (Shanghai, China). Biochemical Technology Co., Ltd. (Shanghai, China). Dimethyl sulfoxide (DMSO) was acquired from Beijing Solarbio Science Technology Co. Ltd. (Beijing, China). High-performance liquid chromatography (HPLC)-grade acetonitrile, methyl tertbutyl ether, formic acid, and ammonium acetate was supplied by Fisher Scientific (Pittsburgh, PA, USA). Ultrapure water was used throughout the study and purchased from Wahaha Group Co., Ltd. (Hangzhou, China). The TRNzol Universal Reagent, FastQuant RT Kit (with DNase), and SuperReal PreMix Plus (SYBR Green) were purchased from Tiangen Biotech Co., Ltd. (Beijing, China).

### 4.2. Instrumentation and Chromatographic Conditions

Analytes were quantified using Ultra-Performance Liquid Chromatography-Tandem Mass Spectrometry (UPLC-MS/MS) which consisted of an LC-30A ultra-performance liquid chromatography (Shimadzu, Kyoto, Japan) and Sciex Triple Quad 5500 tandem triple quadrupole mass spectrometer (AB Sciex, Framingham, MA, USA). Chromatographic separation was performed using an XSelect HSS T3 column (2.1 mm × 100 mm, 2.5 μm, Waters, Milford, MA, USA) at 40 °C by gradient elution. The mobile phase consisted of water with 0.1% formic acid and 5 mM ammonium acetate (phase A) and acetonitrile (phase B), and the gradient elution procedure was as follows: 0–0.5 min, 60–90% B; 0.5–3 min, 90% B; 3–3.1 min, 90–60% B; and 3.1–4 min, 60% B. The injection volume was 5 μL.

The positive ion mode with multi-reaction detection was used. The multiple reaction monitoring transitions of the analytes were m/z 465.3→270.1 for SOR, 468.2→255.4 for SOR-d_3_, 426.1→167.2 for DAPA, and 466.3→195.3 for ^2^H_4_-CAPA (Figure 5). The quantitative parameters, including delustering potential (DP) and collision energy (CE) of the compounds, are summarized in Table 5. Other parameters of the mass spectrometer were as follows: ion source gas 1, 60.0 psi; ion source gas 2, 50.0 psi; curtain gas, 25.0 psi; source temperature, 500 °C; ion spray voltage, 5500 V.

### 4.3. Preparation of Stock Solution and Working Solution

The SOR and DAPA standards were dissolved in dimethyl sulfoxide (DMSO) to make standard stock solutions with final concentrations of 2 mg/mL and 5 mg/mL, respectively. The mixed calibration working solutions were prepared by the dilution of the stock solution with 50% acetonitrile to obtain final concentrations of 50, 150, 500, 2000, 8000, 20,000, 40,000, and 50,000 ng/mL (SOR); 50, 100, 500, 2000, 5000, 10,000, and 20,000 ng/mL (DAPA). Quality control (QC) working solutions with concentrations of 100, 15,000, and 37,500 ng/mL (SOR); 150, 8000, and 15,000 ng/mL (DAPA) were prepared using the same method. A 1 mg/mL stock solution of IS was prepared in DMSO and then diluted with 50% acetonitrile to obtain IS working solution (500 ng/mL of SOR-d_3_ and 500 ng/mL of ^2^H_4_-CAPA). Stock and working solutions were stored at −20 °C and 4 °C, respectively.

### 4.4. Preparation of Calibration Standards and Quality Control (QC) Samples

The calibration standards (CSs) were obtained by spiking 5 μL of the mixed working solution with 45 μL of blank rat plasma. The final concentrations of the calibration curves were 5, 10, 50, 100, 800, 2000, and 5000 ng/mL for SOR and 5, 10, 50, 200, 500, 1000, and 2000 ng/mL for DAPA. QC samples were processed in the same method with the final concentrations of 10, 1500, and 3750 ng/mL for SOR, and 15, 800, and 1500 ng/mL for DAPA.

### 4.5. Plasma Sample Preparation

A volume of 25 µL plasma samples, 2.5 µL IS mixed working solution, and 150 μL methyl tertbutyl ether was added, then vortex-mixed for 1 min. The mixture was centrifuged at 12,000× *g* for 10 min. The organic phase was transferred to a new centrifuged tube and evaporated to dryness under a stream of nitrogen at room temperature. The dried samples were redissolved with 100 μL of 50% acetonitrile and briefly vortexed for 1 min, then 5 μL was injected into the UPLC-MS/MS for analysis.

### 4.6. Method Validation

The method was comprehensively validated according to the guidelines of the Bioanalytical Method Validation Guidance for Industry for the US Food and Drug Administration (US-FDA, 2018) and Chinese Pharmacopoeia (2020). The selectivity, calibration curve, lower limit of quantification (LLOQ), precision, accuracy, matrix effects, recovery, and stability under various conditions were assessed during the method validation process.

#### 4.6.1. Selectivity

The selectivity was evaluated by comparing the chromatograms of blank plasma samples obtained from six batches of rats, blank plasma spiked with analytes at LLOQ and IS, and rat plasma samples. In the absence of interference, the peak area of the analyte in the blank plasma should be less than 20% of the LLOQ and 5% of the IS within the retention time.

#### 4.6.2. Calibration Curve and LLOQ

The calibration curves were validated at 5–5000 ng/mL for SOR and 5–2000 ng/mL for DAPA, respectively. The linearity for each analyte was constructed by the weighted (1/x^2^) least square linear regression of the peak area ratios of analytes to corresponding IS against concentrations. The corrected standard concentration values calculated from the calibration curve were acceptable within ±15% variation of the theoretical values, whereas LLOQ should not exceed 20% to satisfy the requirement.

#### 4.6.3. Precision and Accuracy

The precision and accuracy were evaluated by analyzing QC samples prepared at three concentration levels (low, medium, and high concentration) and LLOQ on three separate days. The precision was calculated as percentage relative standard deviation (RSD, %), and the accuracy was expressed as percentage relative error (RE, %). As required, variation for both precision and accuracy of QC samples were accepted within ±15%, except for LLOQ, which should be within ±20%.

#### 4.6.4. Matrix Effects and Extraction Recovery

The matrix effect for the analyte was determined by comparing the analyte peak area in blank plasma samples from six different batches of rats at three levels of QC samples (*n* = 6 for each level) with the analyte peak area in pure solution. The extraction recovery was assessed by comparing the analyte peak area in extracted plasma samples at three levels of QC samples (*n* = 6 for each level) with the analyte peak area of a blank plasma extract spiked at the same level.

#### 4.6.5. Stability

The stability of QC samples in rat plasma was tested in six replicates at three concentration levels exposed to different storage and processing conditions: autosampler for 6 h after processing, room temperature for 4 h, −80 °C for 30 days, three freeze–thaw cycles (−80 °C to room temperature). The stability of QC samples was concluded to be stable when all analytes did not exceed ±15% of the standard concentration at the three concentrations.

### 4.7. Pharmacokinetic Experiments in Rats

Adult male Sprague-Dawley (SD) rats weighing 230 ± 30 g were provided by Beijing Weitong Lihua Experimental Animal Technology Co., Ltd. (Beijing, China). The license number is SCXK (Beijing) 2016–0011. All experimental procedures for this study were reviewed and approved by the Animal Ethics Committee of Hebei General Hospital (Shijiazhuang, China) (No. 2022072). The rats were adapted to the standard laboratory conditions (12 h dark–light cycle, temperature at approximately 23 ± 2 °C, relative humidity of 50 ± 10%) for a week. All the rats were given access to water freely, while the diet was prohibited for 12 h before administration.

Twenty-four healthy rats were randomly divided into four groups (*n* = 6 in each group). SOR was suspended in 0.5% sodium carboxymethyl cellulose (CMC-Na) with 5% DMSO and DAPA was dissolved in 0.5% CMC-Na. Group 1, the SOR control group, received multiple doses 0.5% sodium carboxymethyl cellulose (CMC-Na) via oral gavage for seven consecutive days and 100 mg/kg SOR via oral gavage on the seventh day, at 1 min after 0.5% CMC-Na administration; Group 2, the SOR combined with multiple-doses DAPA group, received multiple doses 1 mg/kg DAPA via oral gavage for seven consecutive days and 100 mg/kg SOR via oral gavage on the seventh day, at 1 min after DAPA administration; Group 3, the DAPA control group, received multiple doses 0.5% CMC-Na with 5% DMSO via oral gavage for seven consecutive days and 1 mg/kg DAPA via oral gavage on the seventh day, at 1 min after 0.5% CMC-Na with 5% DMSO administration; Group 4, the DAPA combined with multiple-doses SOR, received multiple doses 100 mg/kg SOR via oral gavage for seven consecutive days and 1 mg/kg DAPA via oral gavage on the seventh day, at 1 min after SOR administration. Approximately 0.1 mL of blood samples was collected in centrifuge tubes prefilled with heparin via the oculi choroidal vein at the following time points: 0, 0.5, 1, 2, 3, 4, 5, 6, 7, 8, 10, 12, 24, 36, 48, 72, 96, 120, and 144 h for SOR (Groups 1, 2); 0, 0.25, 0.5, 1, 1.5, 2, 3, 4, 6, 8, 12, 24, and 36 h for DAPA (Groups 3, 4). Blood samples were then centrifuged at 3500× *g* for 10 min at 4 °C, and the supernatant was transferred to another centrifuged tube, then stored in a −80 °C freezer until analysis. Following blood collection, rat liver and intestine tissues were collected for molecular analysis on the seventh day post-treatment with the corresponding drugs for each group. The tissues were stored immediately at −80 °C until used.

### 4.8. Quantitative Real-Time PCR (qRT-PCR) Analysis

The qRT-PCR analysis was used to measure mRNA levels of Ugt1a7 in the liver and intestines of rats. Total RNA was isolated from frozen liver and intestine samples using the TRNzol Universal as per the manufacturer’s instructions. A Bio Tek Epoch (Bio Tek Instruments, Inc., Winooski, VT, USA) was used to quantify the concentration and purity of the total RNA based on the ratio of the absorbance between 260 and 280 nm. RNA samples of 1 μg of were converted to complementary (cDNA) using the FastKing RT Kit. The real-time PCR assays were performed using a two-step amplification method according to the manufacturer’s recommendations in a SLAN-96S Real-Time PCR system (Shanghai Hongshi Medical Technology Co., Ltd. Shanghai, China). The PCR cycling conditions were as follows: 95 °C for 15 min, followed by 40 cycles of 95 °C for 10 s and 60 °C for 32 s. The sequences of primers are listed in Table 6. β-actin was used as an internal control.

### 4.9. Statistical Analysis

DAS 2.1.1 Software (Mathematical Pharmacology Professional Committee of China, Shanghai, China) was used to calculate pharmacokinetic parameters with non-compartmental analysis. The pharmacokinetic analysis included AUC (area under the concentration–time curve), C_max_ (maximum plasma concentration), T_max_ (time to maximum plasma concentration), t_1/2_ (time required to eliminate half of plasma drug concentration), CL_z/F_ (clearance of drug plasma volume per time unit), V_z/F_ (apparent volume of distribution), and MRT (mean residence time). SPSS 25.0 statistical software (SPSS Inc., Chicago, IL, USA) was used to statistically analyze the above-mentioned pharmacokinetic parameters. Statistical comparisons of the experimental and control groups were conducted using a nonparametric rank-sum test or *t*-test, depending on the data type; *p* < 0.05 was considered statistically significant.

## 5. Conclusions

In this study, a rapid and sensitive method for the simultaneous determination of SOR and DAPA in rat plasma samples by UPLC-MS/MS was developed and validated according to the commonly accepted criteria. This method was successfully applied to pharmacokinetic interaction studies between SOR and DAPA. Practical experimentation revealed that oral concomitant administration of SOR with DAPA for multiple doses significantly decreased the oral bioavailability of SOR in rats. In contrast, co-administration of DAPA with multiple doses of SOR inhibited the metabolism of DAPA and increased DAPA bioavailability. These observations suggest pharmacokinetic interactions exist between long-term SOR and DAPA in rats, where Ugt1a7 could play an important role. Therefore, active surveillance for the change of therapeutic effect and adverse reactions should be required when the drugs are combined. However, these interaction studies should be further evaluated by clinical trials.

## Figures and Tables

**Figure 1 molecules-27-06190-f001:**
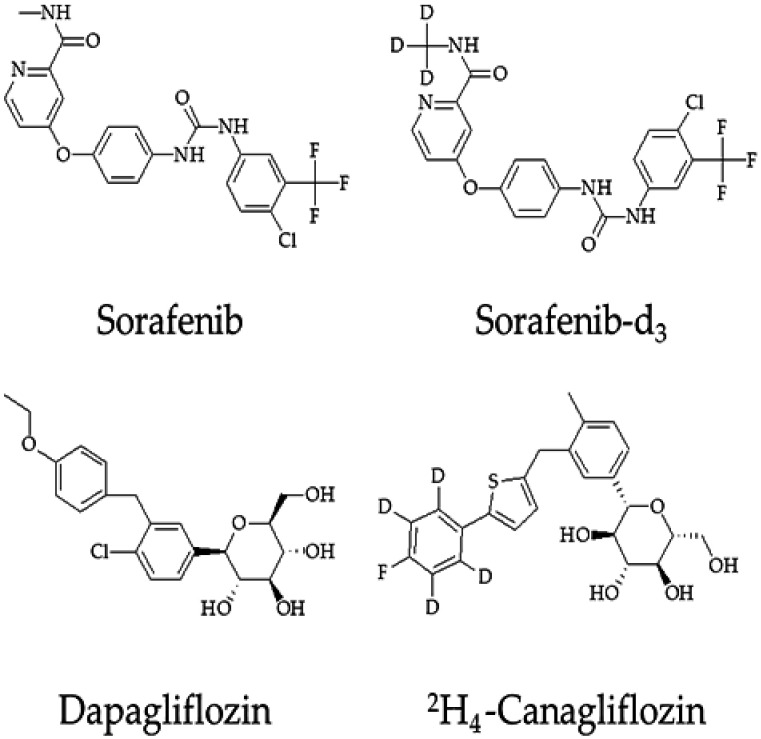
Chemical structure of Sorafenib, Sorafenib-d_3_, Dapagliflozin, and ^2^H_4_-Canagliflozin.

**Figure 2 molecules-27-06190-f002:**
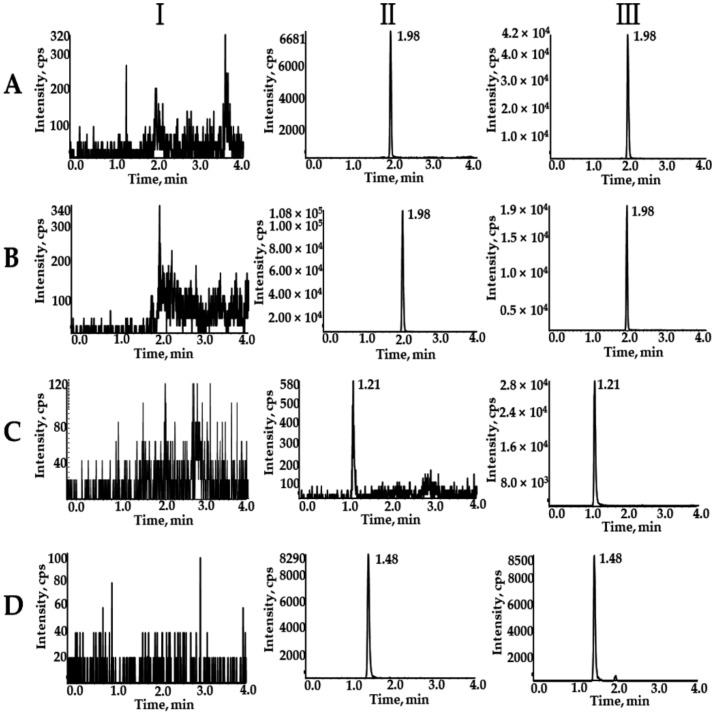
Typical chromatograms of SOR (**A**), SOR-d_3_ (**B**), DAPA (**C**), and ^2^H_4_-CAPA (**D**). Ⅰ, blank plasma; Ⅱ, blank rat plasma spiked with the mixed working solution at LLOQ level and IS; and Ⅲ, rat samples after oral administration of SOR and DAPA.

**Figure 3 molecules-27-06190-f003:**
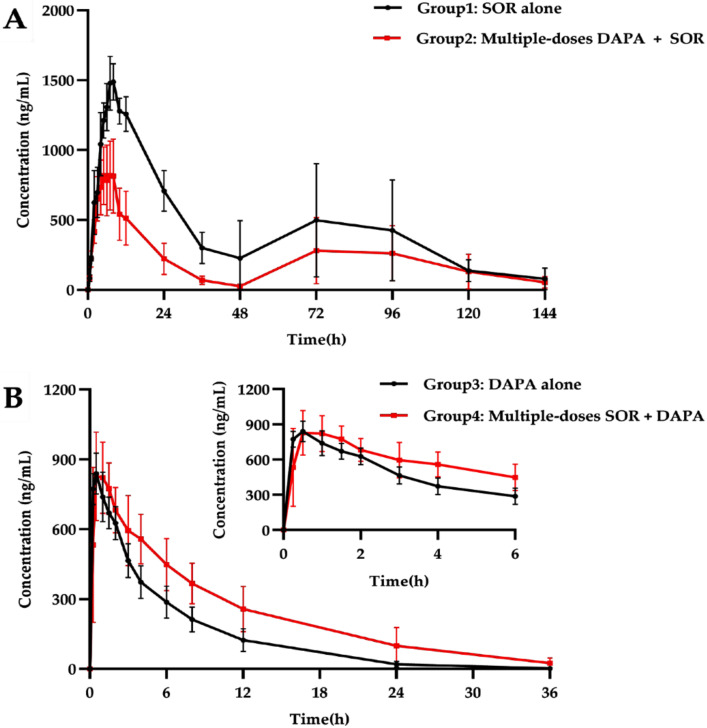
(**A**) The mean plasma concentration–time profiles of SOR after oral SOR alone and combined with multiple-doses DAPA. (**B**) The mean plasma concentration–time profiles of DAPA after oral DAPA alone and combined with multiple-doses SOR.

**Figure 4 molecules-27-06190-f004:**
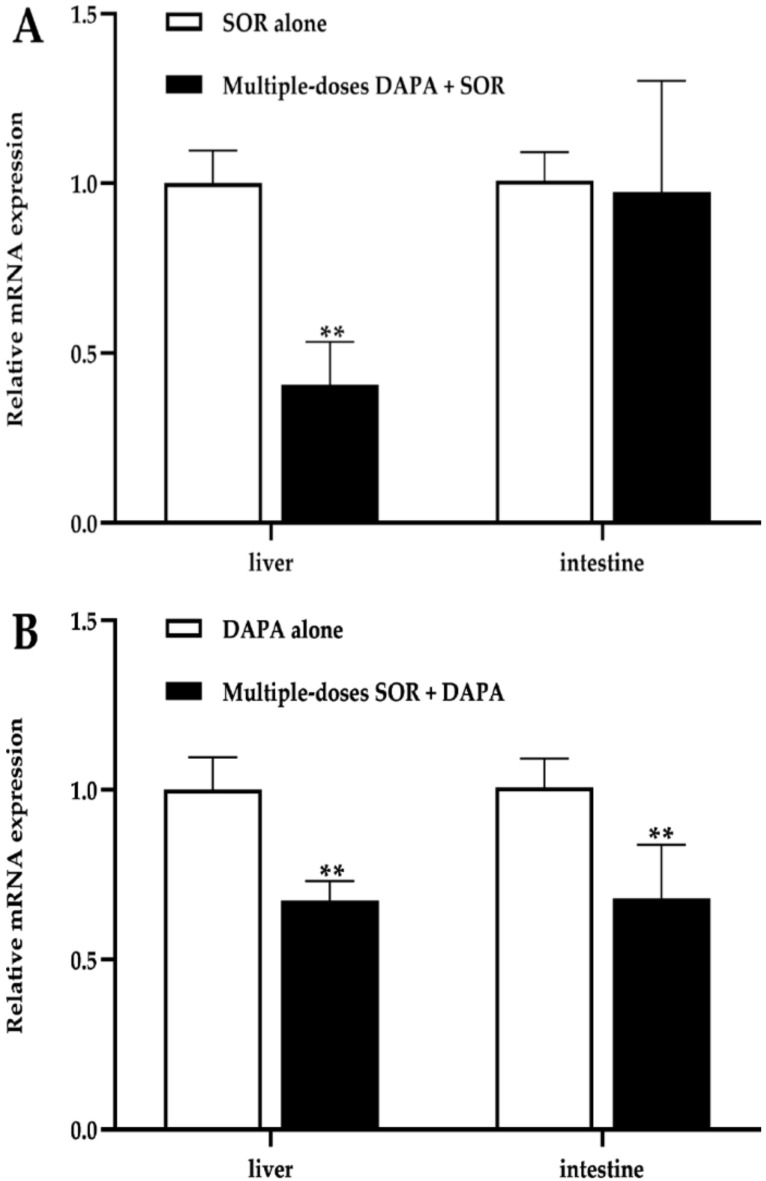
mRNA relative expression ratio of Ugt1a7 in the liver and intestines of rats. (**A**) Effect of multiple-doses DAPA treatment on mRNA expression of Ugt1a7; (**B**) Effect of multiple-doses SOR treatment on mRNA expression of Ugt1a7. ** *p* < 0.01.

**Figure 5 molecules-27-06190-f005:**
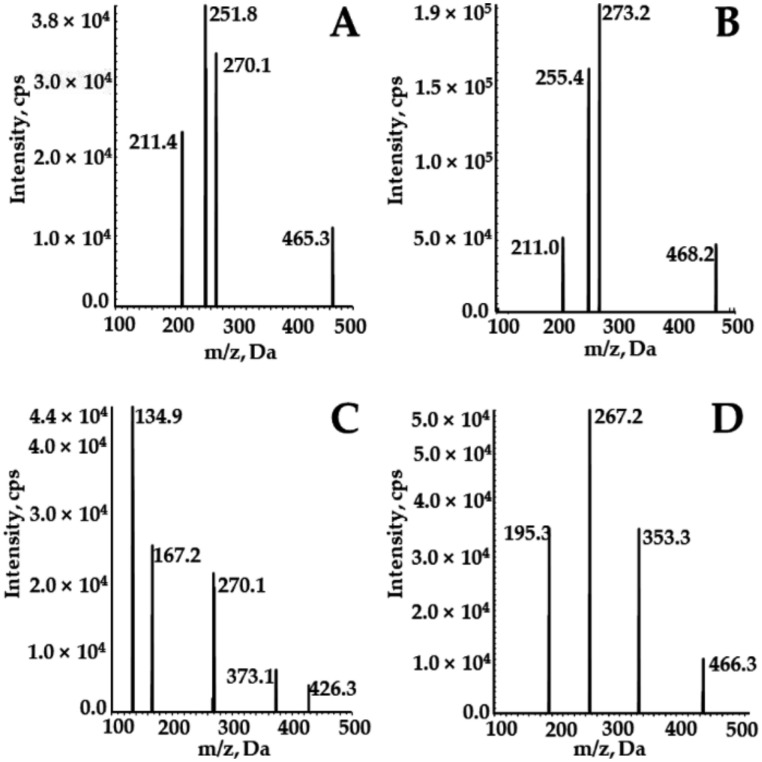
Product ion mass spectrum of SOR (**A**), SOR-d_3_ (**B**), DAPA (**C**), and ^2^H_4_-CAPA (**D**).

**Table 1 molecules-27-06190-t001:** Intra-day and inter-day precision and accuracy of SOR and DAPA in rat plasma.

Analytes	Concentration (ng/mL)	Intra-Day (*n* = 6)			Inter-Day (*n* = 18)		
		Mean ± SD	RSD (%)	RE (%)	Mean ± SD	RSD (%)	RE (%)
SOR	5	5.14 ± 0.27	5.3	2.7	5.03 ± 0.27	5.5	0.6
	10	10.35 ± 0.47	4.6	3.5	10.22 ± 0.51	5.0	2.2
	1500	1538.33 ± 31.89	2.1	2.6	1558.33 ± 82.69	5.3	3.9
	3750	3786.67 ± 77.11	2.0	1.0	3745.56 ± 150.30	4.0	−0.1
DAPA	5	5.16 ± 0.24	4.6	3.2	5.12 ± 0.31	6.1	2.3
	15	15.55 ± 0.80	5.1	3.7	15.42 ± 0.70	4.6	2.8
	800	824.50 ± 14.40	1.8	3.1	815.33 ± 28.31	3.5	1.9
	1500	1546.67 ± 52.03	3.4	3.1	1523.89 ± 95.37	6.3	1.6

**Table 2 molecules-27-06190-t002:** Matrix effects and extraction recovery of SOR and DAPA in rat plasma (*n* = 6).

Analytes	Concentration (ng/mL)	Matrix Effect		Extraction Recovery	
		Mean ± SD (%)	RSD (%)	Mean ± SD (%)	RSD (%)
SOR	10	100.30 ± 3.30	3.3	97.29 ± 3.54	3.6
	1500	96.97 ± 8.80	9.1	106.94 ± 3.90	3.7
	3750	100.78 ± 4.17	4.1	100.41 ± 2.67	2.7
DAPA	15	106.20 ± 5.61	5.3	91.80 ± 2.78	3.0
	800	103.86 ± 4.47	4.3	104.94 ± 4.16	4.0
	1500	101.33 ± 4.65	4.6	105.44 ± 3.44	3.3

**Table 3 molecules-27-06190-t003:** Stability of SOR and DAPA in rat plasma under different conditions (*n* = 6).

Analytes	Conditions	Concentration (ng/mL)	Mean ± SD (ng/mL)	Precision (RSD%)	Accuracy (RE%)
SOR	Autosampler for 6 h	10	10.09 ± 0.53	5.3	0.9
		1500	1538.33 ± 36.56	2.4	2.6
		3750	3716.67 ± 66.23	1.8	−0.9
	Room temperature for 4 h	10	10.68 ± 0.55	5.2	6.8
		1500	1448.33 ± 18.35	1.3	−3.4
		3750	3533.33 ± 66.23	1.9	−5.8
	−80 °C for 30 days	10	10.20 ± 0.39	3.8	2.0
		1500	1533 ± 40.33	2.6	2.2
		3750	3593.33 ± 142.22	4.0	−4.2
	Freeze–thaw stability for three times	10	9.84 ± 0.34	3.4	−1.7
		1500	1550.00 ± 12.65	0.8	3.3
		3750	3601.67 ± 164.00	4.6	−4.0
DAPA	Autosampler for 6 h	15	14.93 ± 0.83	5.5	−0.4
		800	830.00 ± 15.05	1.8	3.8
		1500	1643.33 ± 32.66	2.0	9.6
	Room temperature for 4 h	15	14.90 ± 0.49	3.3	−0.7
		800	889.00 ± 15.43	1.7	11.1
		1500	1653.33 ± 21.60	1.3	10.2
	−80 °C for 30 days	15	15.43 ± 1.07	6.9	2.9
		800	822.17 ± 21.66	2.6	2.8
		1500	1483.33 ± 19.66	1.3	−1.1
	Freeze–thaw stability for three times	15	15.18 ± 0.87	5.7	1.2
		800	769.83 ± 44.56	5.8	−3.8
		1500	1388.33 ± 54.19	3.9	−7.4

**Table 4 molecules-27-06190-t004:** Pharmacokinetic parameters of SOR and DAPA in rats after oral administration alone and combined multiple doses.

Parameters (Unit)	SOR (100 mg/kg)		DAPA (1 mg/kg)	
	Alone	With Multiple-Doses DAPA	Alone	With Multiple-Doses SOR
AUC_0-t_ (μg/L × h)	62,701.33 ± 16,697.65	31,044.48 ± 11,555.87 **	5156.50 ± 1028.25	8572.00 ± 2861.57 **
AUC_0–∞_ (μg/L × h)	63,708.42 ± 17,561.72	34,299.86 ± 12,031.05 **	5180.03 ± 1048.61	9300.01 ± 4261.40 **
C_max_ (μg/L)	1547.87 ± 136.94	904.59 ± 249.56 **	846.00 ± 76.39	870.83 ± 166.87
T_max_ (h)	7.50 ± 0.55	6.00 ± 1.27 *	0.58 ± 0.20	0.83 ± 0.41 **
t_1/2z_ (h)	17.456 ± 5.65	24.82 ± 20.13	4.51 ± 0.67	7.95 ± 4.03 **
CL_z/F_ (L/h/kg)	1.69 ± 0.52	3.35 ± 1.52 *	0.20 ± 0.04	0.12 ± 0.03 **
V_z/F_ (L/kg)	37.43 ± 9.78	106.53 ± 69.61 *	1.27 ± 0.19	1.12 ± 0.10
MRT_0-t_ (h)	47.08 ± 9.48	54.99 ± 13.07	6.10 ± 0.96	9.01 ± 1.22 **
MRT_0–∞_ (h)	48.73 ± 10.48	67.64 ± 33.94	6.25 ± 1.07	11.22 ± 4.86 **

* *p* < 0.05, ** *p* < 0.01, compared with SOR or DAPA alone, indicating statistically significant difference. The main pharmacokinetic parameters are shown as the mean ± standard deviation.

**Table 5 molecules-27-06190-t005:** Experimental setting for the tandem mass-spectrometer for the analytes and internal standards.

Experimental Setting	SOR	DAPA	SOR-d_3_	^2^H_4_-DAPA
MRM transition	465.3→270.1	426.1→167.2	468.2→255.4	466.3→195.3
Delustering potential (DP), V	100	80	100	100
Collision energy (CE), V	45	30	45	45
Collision cell exit potential (CXP), V	7	7	7	7
Entrance potential (EP), V	10	10	10	10

**Table 6 molecules-27-06190-t006:** Primers sequences for qRT-PCR analysis.

Gene	Forward Primer	Reverse Primer
Ugt1a7	5′ AGTGTCCGTTTGGTTGTT-3′	5′-TTCCATCGCTTTCTTCTC-3′
NAPDH	5′-GCCTTCCGTGTTCCTACC-3′	5′-GCCTGCTTCACCACCTTC-3′

## Data Availability

Not applicable.

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
