# Peer review of "Development of UPLC-MS/MS Method to Study the Pharmacokinetic Interaction between Sorafenib and Dapagliflozin in Rats"

_molecules, 2022, doi:10.3390/molecules27196190_

Round 1

Reviewer 1 Report

The authors validated a UPLC-MS/MS method for the determination of SOR and DAPA in rat plasma to evaluate the effect of co-administration of these two drugs. The results of the study identified that pharmacokinetic interactions were present, so the coadministration of these two drugs should be evaluated. 

The research work is well supported, the summary is adequate, it allows understanding the context of the research, the objective of the work is evidenced, the most important results are reported and it concludes in an adequate manner. The introduction is adequate, the analysis of the results is theoretically supported, and it gives solidity to the authors' opinions. 

Regarding the validation of the method, it would be important to describe the treatment of the rats, the method of obtaining the plasma and the handling conditions of the plasma after its collection.

Reviewer 2 Report

The authors presented the research work entitled "Development of UPLC-MS/MS Method for Studying the Phar- 2 macokinetic Interaction Between Sorafenib and Dapagliflozin 3 in Rats" nicely and descriptive way. However, there are major issues associated with the manuscript as follows:

1. A significant number of hypotheses are described for the possible mechanism behind the pharmacokinetic interaction between sorafenib and dapagliflozin with necessary support from the literature like the involvement of UGT, plasma protein, CYP. Authors finally stated that UGT1A9 is responsible to cause drug interaction. Authors have to perform a minimum of one WB or RT-PCR-based study to ascertain the main role of UGT1A9, which is expected to be the main culprit. In the absence of suitable mechanistic data, the manuscript is worthless to publish in the present journal, which publishes quality information and maintains a high standard in this field.

2. To justify the pharmacokinetic interaction data, please remove the canagliflozin interaction information, which is unpublished to date, as relevant reference(s) is enough to indicate this particular case.

Reviewer 3 Report

This manuscript describes a bioanalytical method “UPLC-MS/MS” preferably used for determination of active constituents and their metabolites in plasma during drug development. The work presented in this method article is interesting, and should be considered for publication, but at this point, few issues need to be addressed before publication can be recommended.

Specific issues are detailed below;

1.       In the title, its more appropriate to use “to study” rather than “for studying”.

2.       Rephrase the line from 354-359, as the sentences are too long and seems to be complicated.

3.       All the symbols (example; ng/mL) should be unified throughout the manuscript. For example, line 335 and 336.

4.       There are few grammatical and spelling mistakes, must be corrected.

5.       SOR was suspended in 0.5% sodium carboxymethyl cellulose (CMC-Na) with 5% DMSO. How, the researcher evaluated the safety of administered dose in rats.

Round 2

Reviewer 2 Report

I appreciate for satisfactory revision of the manuscript by the authors. The only thing that is lacking and has to be justified in the revised MS is as follows:  One case is the presence of dapagliflozin decreases the sorafenib plasma exposure and increases the clearance. Another case is the presence of sorafenib increases the dapagliflozin plasma exposure and decreases the clearance. In both cases, mRNA expressions decrease in the liver.  Authors are requested to explain with suitable examples in the 'Discussion' section of revised MS. 
